# Electrostatic interactions guide substrate recognition of the prokaryotic ubiquitin-like protein ligase PafA

Matthias F. Block [1], Cyrille L. Delley [1,2], Lena M. L. Keller [1], Timo T. Stuehlinger[1] & Eilika Weber-Ban [1] ✉

Pupylation, a post-translational modification found in *Mycobacterium tuberculosis* and other Actinobacteria, involves the covalent attachment of prokaryotic ubiquitin-like protein (Pup) to lysines on target proteins by the ligase PafA (proteasome accessory factor A). Pupylated proteins, like ubiquitinated proteins in eukaryotes, are recruited for proteasomal degradation. Proteomic studies suggest that hundreds of potential pupylation targets are modified by the sole existing ligase PafA. This raises intriguing questions regarding the selectivity of this enzyme towards a diverse range of substrates. Here, we show that the availability of surface lysines alone is not sufficient for interaction between PafA and target proteins. By identifying the interacting residues at the pupylation site, we demonstrate that PafA recognizes authentic substrates via a structural recognition motif centered around exposed lysines. Through a combination of computational analysis, examination of available structures and pupylated proteomes, and biochemical experiments, we elucidate the mechanism by which PafA achieves recognition of a wide array of substrates while retaining selective protein turnover.

Regulated protein degradation, mediated through chaperone-proteases, is a fundamental process across all forms of life. While bacteria utilize the Clp proteases, archaea and eukaryotes rely on proteasomes. Notably, in addition to the conventional bacterial Clp protease complexes, most Actinobacteria possess a proteasome assembly that contributes to their survival during demanding circumstances[1,2]. For instance, the proteasome was shown to be essential for the human pathogen *Mycobacterium tuberculosis* (Mtb) to persist in the lungs of infected mice[3], and proteasomal degradation also plays a pivotal role in controlling the mycobacterial DNA damage response[4,5].

To ensure controlled protein turnover, organisms have developed dedicated mechanisms for substrate recruitment. In eukaryotes, this involves the attachment of the small protein ubiquitin (Ub) to lysine side chains, thereby marking proteins for proteasomal degradation[6]. In Actinobacteria, a functionally similar but enzymatically distinct pathway called pupylation has evolved. This process is mediated by a single ligase, known as proteasome accessory factor A (PafA), which catalyzes the formation of an isopeptide bond between a substrate lysine residue and the C-terminal glutamate of the prokaryotic ubiquitin-like protein (Pup). During this ligation reaction, PafA first phosphorylates the C-terminal glutamate side chain of Pup using ATP. Subsequently, a nucleophilic substitution occurs, with the ε-amino group of a substrate lysine attacking the mixed anhydride at the carbonyl carbon atom, resulting in the formation of an isopeptide bond between the substrate protein and Pup[7,8]. Interestingly, mycobacteria encode Pup in a ligation-incompetent form, ending with a C-terminal glutamine (PupQ). To generate the ligation-competent PupE species, PupQ must undergo deamidation through the action of the deamidase of Pup (Dop)[7,9,10]. Pupylated proteins are recognized by the proteasome-associated ATPase (Mpa in mycobacteria; ARC in other actinobacteria) and subsequently translocated into the 20S proteasome[11–14]. Alternatively, pupylation can be reversed by Dop, which also can act as

[1]ETH Zurich, Institute of Molecular Biology & Biophysics, Zurich, Switzerland. [2]Present address: University of California, San Francisco, USA.
✉e-mail: eilika@mol.biol.ethz.ch

a depupylase[15,16]. The deamidation process is chemically equivalent to depupylation, as both reactions result in the cleavage of the C-N bond. However, deamidation releases ammonia, whereas depupylation releases the modified lysine from Pup.

PafA and Dop are close structural homologs likely originating from gene duplication, which have evolved to catalyze opposing reactions, namely C−N bond formation and hydrolysis[7,17]. While the structural features and the reaction mechanisms of these enzymes have been studied, the basis for substrate selection remains poorly understood. In the ubiquitination pathway, a cascade of enzymes consisting of activating E1, conjugating E2, and ligating E3 enzymes are responsible for binding and transferring Ub, recognizing substrates, and covalently attaching Ub to specific targets. Hundreds of E3 ligases confer specificity and allow regulation of substrate ubiquitination[6]. In contrast, PafA carries out all steps required for pupylation, and it serves as the sole Pup ligase in the pupylation pathway. Disruption of the *pafA* gene abolishes all pupylation activity[14].

Substrate selectivity is thought to arise, in part, through discriminatory affinities of pupylated substrates towards Dop, in contrast to indiscriminate recruitment to the proteasome via its gate keeper Mpa[18,19]. Several research groups have independently published mass spectrometry data sets of whole-lysate, affinity-purified pupylated proteins (referred to as pupylomes) from mycobacteria and other actinobacteria[20–24]. These studies identified hundreds of putative pupylation substrates, spanning proteins from diverse functional classes without a clear preference. Bioinformatic analysis of these pupylomes has revealed statistical differences in amino acid frequencies in the sequence neighborhood of target lysine residues compared to non-target lysines[25–27]. However, these patterns are insufficient to predict experimentally validated target lysines.

Studies involving the transplantation of the pupylation machinery to *Escherichia coli*[28] or its utilization as a proximity-labeling method in eukaryotes[29], have demonstrated the pupylation of a number of proteins that would never encounter the PafA ligase in their natural environment. These findings might indicate that PafA indiscriminately pupylates proteins that present surface accessible lysine residues. However, functional connections to proteasomal degradation or an increase in steady-state levels have only been observed for a limited number of substrates in pupylation-deficient or proteasome knockout strains[30,31]. Collectively, these observations suggest that PafA exhibits selectivity towards its substrates, although the molecular recognition mechanism remains unknown.

By conducting biochemical experiments and mutational studies on the ligase PafA and its genuine substrate FabD (malonyl CoA-acyl carrier protein transacylase), we demonstrate that PafA exhibits specificity towards its substrates and reveal how this specificity is encoded. Our findings indicate that PafA selects its substrates based on tertiary structure features rather than linear sequence motifs or characteristics. Through careful interaction studies, we show that selective binding between PafA and its protein substrate is mediated by only a few electrostatic interactions, resulting in a relatively small binding interface. The compact nature of the interface allows for rapid evolutionary adaption of the bacterial proteasome system to new substrates.

## Results

### Pupylome data are biased by protein abundance

During in vitro pupylation experiments on pupylation targets identified in mycobacterial pupylome studies, we and others[14,32–34] observed a significant discrepancy in the pupylatability among different reported pupylome members. For example, when subjected to the same reaction conditions and at the same concentrations, we were able to achieve complete modification of PanB (ketopantoate hydroxymethyltransferase) with Pup within 2 h, while Adk (adenosine kinase), another member of the pupylome, required a much longer period of 20 h to attain full pupylation (Fig. 1a). Furthermore, certain pupylome

members such as MDH (malate dehydrogenase) or ClpP2 (ATP-dependent Clp protease subunit 2) exhibited only marginal or negligible pupylation levels under the same reaction conditions (Fig. 1a).

Notably, Mtb proteins like FabD or PanB, for which increased steady state levels were detected in pupylation- or proteasome-deficient strains[30], are pupylated efficiently and quantitatively in vitro[8,32] (Supplementary Fig. 1a). This suggests that a significant fraction of these proteins undergo pupylation and subsequent turnover in vivo. In contrast, pupylome members that cannot be pupylated in vitro are likely to be either false positives or proteins for which only a minor fraction within the cell undergoes pupylation.

All pupylome studies conducted thus far employed affinity enrichment methodology with affinity-tagged Pup, and mass spectrometry was employed to identify proteins within the enriched pool. While this approach enables the identification of proteins carrying the modification, it lacks quantitative information regarding the overall abundance or the ratio of modified to unmodified species for a given protein. Since the cellular concentrations of different Mtb proteins have been shown to vary greatly, spanning at least four orders of magnitude[35], we examined whether the reported pupylome data exhibit a bias towards more abundant proteins. To assess this, we looked for a correlation between protein abundance and the presence of a given protein in the determined pupylomes (Fig. 1b). We also investigated potential correlations with molecular weight (Fig. 1b) or the number of lysines in a protein (Supplementary Fig. 1b), as it had previously been suggested that larger proteins are pupylated preferentially[36]. Additionally, larger proteins tend to have more lysine residues on average, which results in the generation of more peptides during tryptic digestion, favoring both affinity enrichment and detection. We observed a clear bias towards higher protein abundance in the proteins reported in the pupylome—on average, they were 4-fold more abundant (Welch's *t*-test p-value $6.4 \times 10^{-9}$, Kolmogorov-Smirnov test $p$-value $1.6 \times 10^{-55}$), whereas a bias in molecular weight was less pronounced, with an average weight difference of 18% or 7.9 kDa (Welch's *t*-test p-value $4.5 \times 10^{-7}$, Kolmogorov-Smirnov test $p$-value $4.2 \times 10^{-11}$) (Fig. 1b). Similarly, we detected 1.5 times more lysine residues in pupylation substrates (Welch's *t*-test p-value $3.1 \times 10^{-18}$, Kolmogorov-Smirnov test $p$-value $1.7 \times 10^{-17}$) or a 1.2-fold higher lysine density (Welch's *t*-test $p$-value $2.8 \times 10^{-10}$, Kolmogorov-Smirnov test $p$-value $1.0 \times 10^{-10}$) (Supplementary Fig. 1b).

We then classified different substrates as good (+++), intermediate (++), or bad (+) based on experiments presented in this study or in the literature (see Supplementary Table 1). Good substrates are efficiently pupylated in vitro and/or have been shown to accumulate in Pup-proteasome system (PPS) member knockout studies. For intermediate substrates, pupylation can be observed in vitro, but at a slower rate and to a lower extent compared to good substrates, but they may still show effects on viability in vivo. Bad substrates are defined as substrates that are not significantly pupylated in in vitro experiments but were nevertheless identified in pupylome studies. We compared the proteins from these three categories with their position in the proteome abundance or molecular weight distribution plot (Fig. 1c, individual substrates are listed in Supplementary Table 1). Despite the naturally smaller sample size, this analysis supports the tendency for high abundance proteins to be represented in the pupylomes, while showing an insignificant correlation with molecular weight. Furthermore, good pupylation substrates are typically reported in more than one pupylome study (larger circle), but rarely in all three, highlighting considerable stochasticity in the proteomic detection. Poor pupylation substrates and good pupylation substrates are found in all mycobacterial pupylome studies (Fig. 1c). In contrast, there is a strong correlation between in vitro reactivity and steady-state accumulation in vivo (lime green circle) in pupylation-deficient strains[30]. Pupylation substrates shown to accumulate in vivo, such as Mpa[30,32], FabD[30], PanB[30], Ino1[21], Icl1[21], and RecA[37], also exhibit high reactivity in vitro. This suggests that the likelihood of detecting a

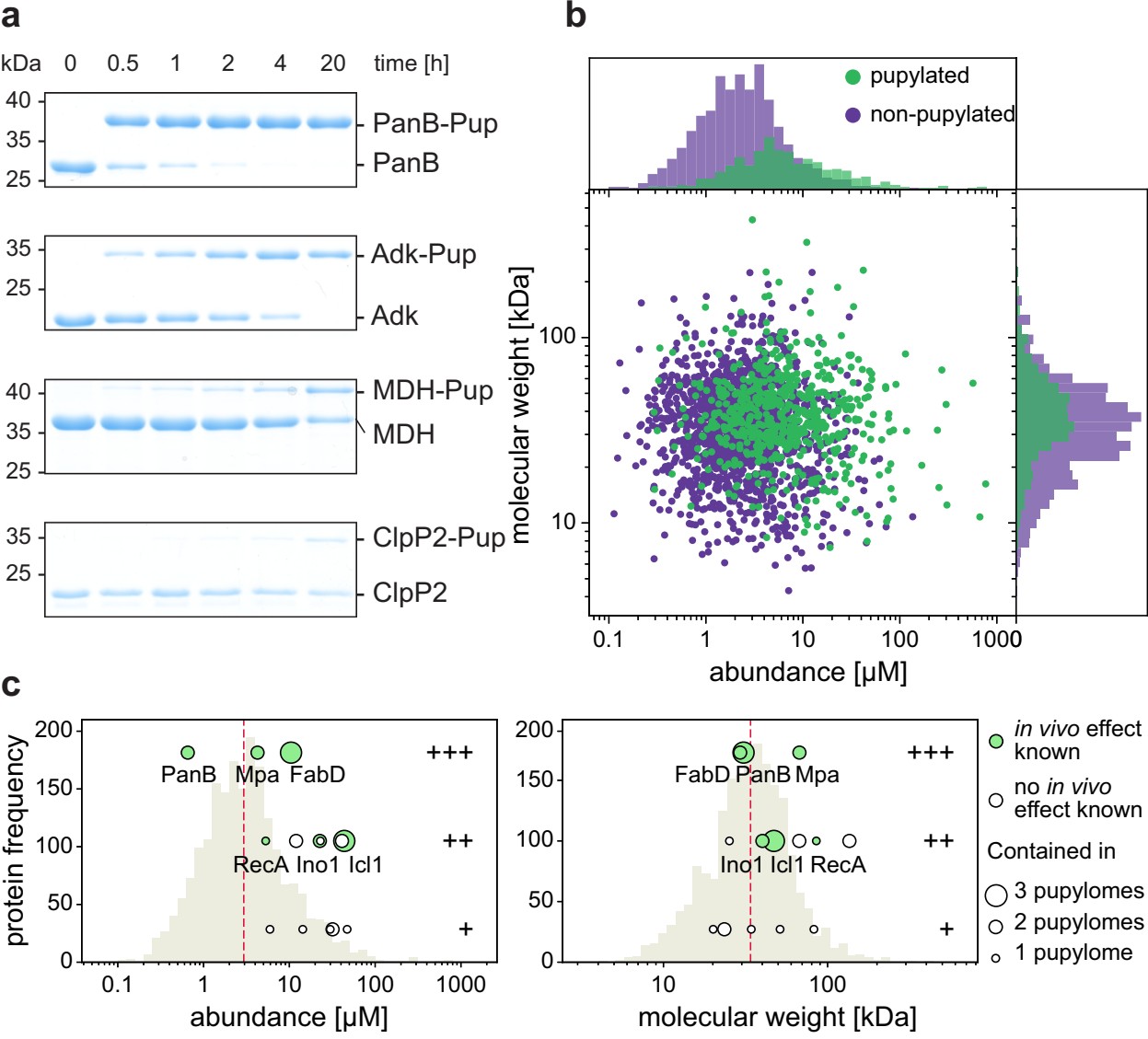

**Fig. 1 | Pupylomes are biased by protein size and abundance. a** In vitro pupylation of four different substrates followed by polyacrylamide gel electrophoresis (PAGE) and Coomassie-staining. All four substrates were identified as pupylation substrates in pupylome studies but demonstrate different reactivity in vitro. Representative gels of three individual experiments are shown. Source Data (uncropped gels) are provided as a Source Data file. **b** Protein abundance as determined in a previous mass spectrometry study[35] is plotted against protein molecular weight for members of the Mtb proteome. Green circles represent proteins that have been identified at least once as pupylated in one of the three mycobacterial pupylomes. Purple circles represent non-pupylated proteins in mycobacteria. **c** Proteins that were reconstituted and pupylated in vitro are depicted as circles and plotted versus abundance (left) and molecular weight

(right). The histogram in the background (beige) depicts the whole proteome with the red line marking the median of non-pupylated substrates. Substrates were categorized into good (+++), intermediate (++) and bad (+) substrates. Good substrates are efficiently pupylated in vitro and/or have been shown to accumulate in PPS knockout studies. For intermediate substrates, slower pupylation is observed in vitro and to a lower extent compared to good substrates. Bad substrates, although identified in pupylome studies, are not significantly pupylated in vitro. A full list of substrates can be found in Supplementary Table 1. The size of the circle represents the number of mycobacterial pupylome studies that identified the protein among the substrates. Lime green circles depict proteins with known physiological effect caused by pupylation, the individual proteins are labeled. Where no effect is known, open circles are shown.

protein in the pupylome is more closely related to the absolute number of pupylated molecules rather than the relative degree of pupylation. Consequently, the detection of a protein as a member of the pupylome has limited predictive power regarding the biological significance of its pupylation or the efficiency of its pupylation by PafA.

### Three-dimensional context of target lysines is required for efficient pupylation

Because of the low predictive power of the pupylome for protein substrate reactivity in the PafA-catalyzed ligation, we investigated the characteristics of known target lysines from pupylation substrates with

good reactivity. We aligned the amino acid sequences ranging from ten residues upstream to 10 residues downstream of the pupylation target lysines in order to discover putative sequence patterns present in good and intermediate PafA substrates (see Supplementary Table 1). However, the alignment did not reveal any obvious recognition motif encoded in the primary structure (Supplementary Fig. 1c).

To assess whether steric constraints govern pupylation, we used the IUPred3 web server implementation[38] to predict disorder in several substrates for which we possessed information regarding the pupylation site and the corresponding in vitro pupylatability. All identified pupylation sites were found within predicted structured regions of the

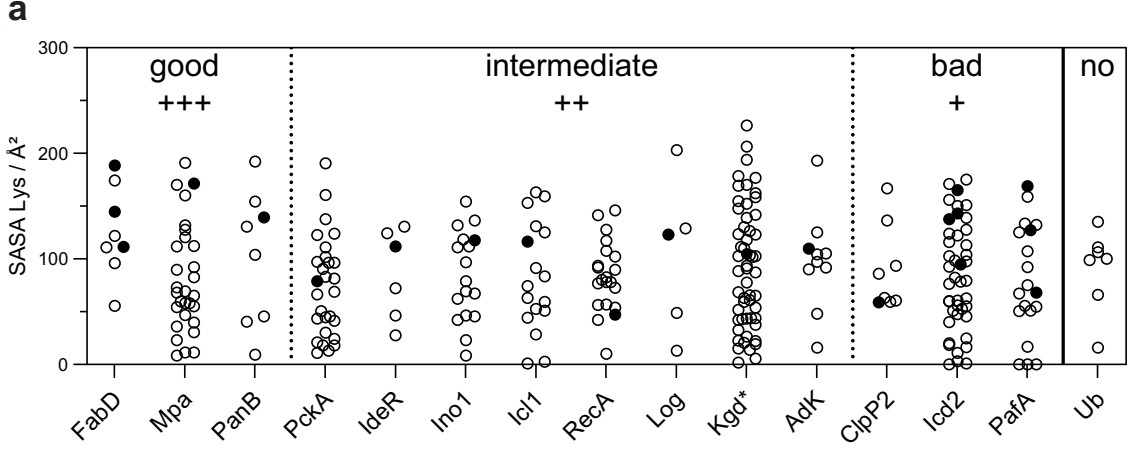

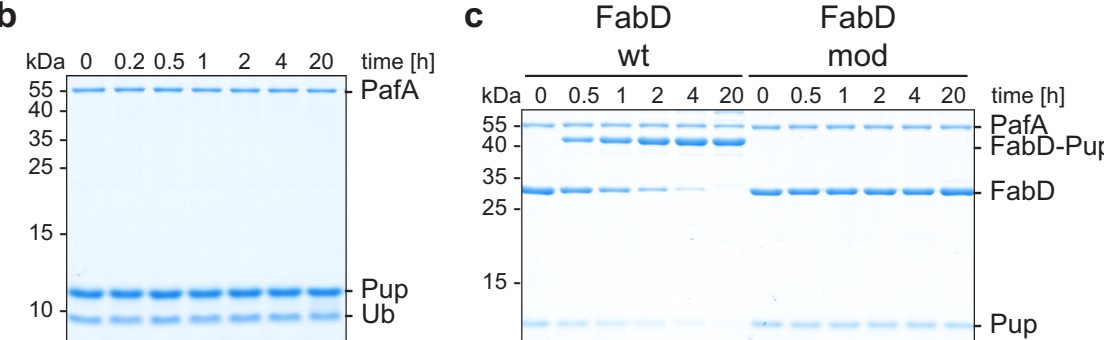

**Fig. 2 | Pupylation specificity of certain lysine residues is not encoded in their accessibility or linear sequence context. a** Dot plot depicting the solvent accessible surface area (SASA) of lysine residues present in different pupylome members. The lysine SASA is based on available crystal structures or AlphaFold2 predictions (*) (see Supplementary Table 1). A filled circle indicates the reactive lysine identified in pupylome studies or determined from purified pupylated substrates. All other lysines are shown as open circles. Good (+++) substrates are efficiently pupylated in vitro and/or have been shown to accumulate in PPS knockout studies. For intermediate substrates (++), slower pupylation is observed in vitro and to a lower extent compared to good substrates. Bad substrates (+), although identified in pupylome studies, are not significantly pupylated in vitro. For further information see Supplementary Table 1. Ubiquitin (Ub) shows no in vitro reactivity. **b** Pupylation of Ub followed by SDS-PAGE and Coomassie-staining. No pupylation was detected within 20 h after starting the reaction. Representative gels of three individual experiments are shown. Source Data (uncropped gels) are provided as a Source Data file. **c** Pupylation time courses of native and S-carbamidomethylated (mod) FabD followed by SDS-PAGE and Coomassie-staining. While the native variant of FabD was almost completely pupylated within 4 h, modified FabD remained unpupylated over the course of 20 h. Representative gels of three individual experiments are shown. Source Data (uncropped gels) are provided as a Source Data file.

respective proteins (Supplementary Fig. 2), except for K229 of IdeR, the penultimate residue, which was predicted to be disordered. However, based on published structures of IdeR[39] and the AlphaFold prediction (pLLD > 90), this lysine is part of a structured region.

Next, we examined the solvent-accessible surface area (SASA) of targeted and non-targeted lysine side chains in pupylome members with available 3D structures (Fig. 2a). The analysis revealed that lysine residues targeted for pupylation exhibit an average accessibility, characteristic of surface-exposed residues that are not situated in flexible loops. Consequently, the solvent accessibility of targeted lysine side chains (filled circle) did not exhibit any systematic differences compared to lysines that are not pupylated (open circle). Furthermore, we did not observe a significant disparity between target lysines in substrates that are pupylated to a high degree and those that are pupylated to an intermediate or lower degree. This indicates that accessibility alone cannot account for the preference of PafA for specific lysine residues or predict the pupylatability of different substrates.

To further validate that PafA exhibits selectivity towards genuine substrates beyond the surface-accessibility of lysines, we investigated the pupylation of the non-substrate Ub. Inherent to its cellular function as regulatory and signaling protein modification in eukaryotes, Ub possesses seven surface-exposed lysines that can be ubiquitinated in

order to build different poly-Ub chains[6]. Although these seven lysine residues exhibit similar solvent-accessible surface areas as genuine pupylation substrates (Fig. 2a) and are modified during Ub chain formation, Ub cannot be pupylated under the same conditions in which genuine substrates are readily pupylated (Fig. 2b).

To distinguish between recognition of structural motifs and linear sequence motifs, we generated structurally perturbed variants of FabD and PanB by S-carbamidomethylation. Treating previously buried cysteine residues of a protein with iodoacetamide in the unfolded state can perturb protein refolding when returning to non-denaturing conditions. Circular dichroism (CD) spectra of FabD and PanB revealed alterations in secondary structure compared to the native protein, indicating structural disorder induced by S-carbamidomethylation (Supplementary Fig. 1d). Modified FabD (Fig. 2c) and PanB (Supplementary Fig. 1e) did not undergo pupylation within 20 h, whereas the respective native proteins were fully pupylated within 4 h. Since cysteines in both proteins are not in the immediate vicinity of the targeted lysine but at a distance of ≥ 20 Å between the C-alphas, steric obstruction of the pupylation site can be ruled out as a reason for the disruption of protein pupylation. This suggests a substrate recognition mechanism where selectivity of PafA for its protein substrates is encoded in the tertiary structure rather than the linear amino acid

sequence surrounding the targeted lysine. Furthermore, the example of Ub demonstrates that surface-exposed lysines are not indiscriminately recognized and pupylated by PafA, indicating the existence of selection determinants beyond mere accessibility.

## Key amino acids surrounding the target lysine in FabD are required for recognition by PafA

For several pupylation substrates with known in vitro pupylatability, including FabD, molecular structures are available (FabD: 2QC3; Mpa: 7PXC; PanB: 1OY0; PckA: 4R43; IdeR: 1U8R; Ino1: 1GR0; Icl1: 1F8I; RecA: 4PTL; Adk: 2CDN; ClpP2: 5EOS; Icd2: 4KVU). For substrates without available structures, AlphaFold predictions[40,41] were used (Log: AF-O05306; Kgd: AF-P9WIS5). Analysis of the protein surface surrounding the pupylation target lysines in these substrates reveals a characteristic protein-protein binding interface[42] with a hydrophobic core surrounding the lysine and a charged perimeter within a roughly 15 Å radius of the lysine's C-alpha atom (Supplementary Fig. 3). Notably, among the three different substrate categories ("good", "intermediate" and "bad"), only the last category exhibits a difference in the surrounding environment of the pupylation site. For example, in the case of ClpP2, the lysine is either less accessible or surrounded by fewer charged residues compared to the other substrates. To investigate the interaction determinants in this proximal surface of the target lysine, we experimentally probed the main pupylation target lysine of FabD. To focus on the main pupylation site, we used a FabD variant that lacks secondary pupylation sites, as previously described[43]. For simplicity, in the following, this variant will be referred to as FabD. The primary pupylation site in FabD, K173[43], resides in the smaller, non-catalytic domain of FabD, which adopts a ferredoxin-like fold[44]. The potential interaction interface is formed by α-helix 8 and α-helix 9, which run antiparallel to each other and are packed against a four-stranded β-sheet that is not part of the interaction interface itself (Fig. 3a).

To assess the contributions of individual residues, we conducted an alanine scan of surface-exposed residues surrounding K173. Each surface-exposed residue within a 20 Å radius of the target lysine was individually mutated to alanine, resulting in eleven FabD alanine variants (Fig. 3a and Supplementary Fig. 4a). In vitro pupylation assays were performed for each of these constructs, monitoring product formation over a 20-h time course using SDS-polyacrylamide gel electrophoresis (PAGE) and Coomassie-staining. Varying levels of attenuation of the progression of the reaction were observed for different residues surrounding the target lysine. These ranged from full reactivity to negligible pupylation within 4 h under conditions where FabD is typically completely pupylated (Fig. 3b and Supplementary Fig. 4b). Certain specific residues demonstrated a particularly strong impact on pupylation, including the positively charged R145, as well as the negatively charged residues D150, E176 and D177 (Fig. 3b). Importantly, the CD spectra of the alanine variants were identical to the CD spectrum of FabD, indicating that the mutations did not affect the overall fold of the protein (Supplementary Fig. 4c). Substituting other positive, negative, polar, or hydrophobic residues with alanine had considerably milder effects on reactivity (Supplementary Fig. 4a). Importantly, all the residues that impair pupylation are located within the potential three-dimensional interaction surface surrounding the target lysine. In the linear amino acid sequence, only some of these residues are located on the same helix in close proximity in sequence to the lysine, while others are part of α9 and located at least 16 residues away from the lysine.

## Conserved residues surrounding the PafA active site mediate substrate recognition

In this study, PafA from *Corynebacterium glutamicum* (Cgl) was chosen for its superior stability and ease of expression and purification. Furthermore, the structure of CglPafA has been determined[45,46], allowing

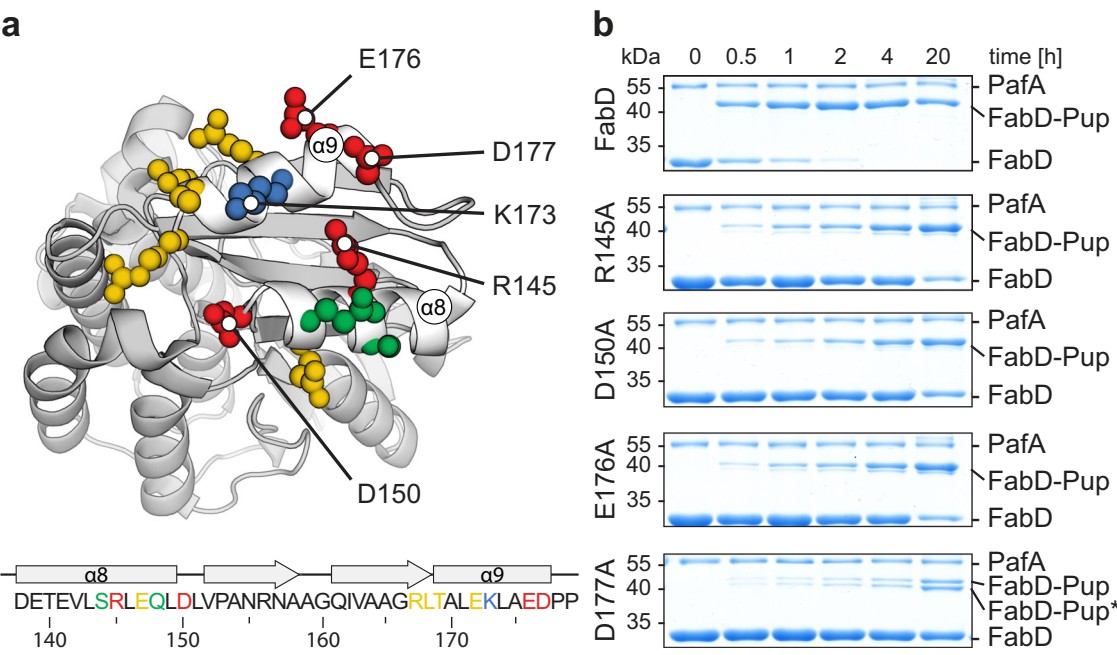

**Fig. 3 | The pupylation motif on FabD is conformational and not linear.**
**a** Cartoon representation of the FabD structure (PDB: 2QC3). Residues around K173 (blue) are colored according to their contribution to the pupylation reactivity of FabD as judged by the effect of the mutation to alanine. Red color indicates a strong contribution manifested as a severe pupylation defect upon mutation. Green colored residues had little effect in the alanine scan and yellow colored residues showed an intermediate effect. **b** Pupylation time courses recorded by SDS-PAGE and Coomassie-staining for those FabD alanine variants that had the strongest

effect on pupylation efficiency. Pupylation time courses for other variants are shown in Supplementary Fig. 4b. Based on these pupylation assays, contribution of individual residues to pupylation efficiency was mapped in **a**. The band labeled FabD-Pup* was identified by mass spectrometry as FabD pupylated at residues K228 or K291 instead of the native pupylation site K173 (see Supplementary Data 2). Representative gels of three individual experiments are shown. Source Data (uncropped gels) are provided as a Source Data file.

for assessment of individual residue positions in a three-dimensional context. To prevent self-pupylation of [Cgl]PafA at lysines K106, K172 and K302, which were identified as pupylation targets through mass spectrometry (Supplementary Data 1), these residues were mutated to arginine. As they are located on the "back" side of PafA, away from the active site, they are not expected to affect the pupylation reaction. Consistent with this assumption, no differences in substrate pupylation were observed between the variant and wild type [Cgl]PafA (Supplementary Fig. 5a). Therefore, throughout the in vitro experiments, [Cgl]PafA_{K106R, K172R, K302R} was used unless otherwise stated, and it is referred to as PafA due to its behavior closely resembling that of the wild type.

The active site of PafA is located within a highly conserved β-sheet consisting of six antiparallel β-strands. Flanking the active site are several loops that form a rim around the shallow, roughly 40–50 Å wide concave β-sheet cradle[46]. Positioned at one side of this cradle is the ATP-binding site, while the C-terminal glutamate of Pup enters from the opposite direction, allowing for formation of the phospho-Pup intermediate in the active site[8]. The active site, as well as several residues making up the rim, are highly conserved in Actinobacteria and most of these rim residues are also conserved between mycobacteria and corynebacteria (Supplementary Fig. 5b and 5c).

Within the rim, we identified three small clusters of positively charged residues (Fig. 4a), which could in principle interact with the charged residues located within the 15 Å perimeter of the target lysines. These clusters were designated as site 1 to 3 based on their position in the linear sequence. Site 1, positioned adjacent to the Pup binding groove at one end of the cradle, consists of two arginines (R32 and R38). Site 2 and 3, situated towards the opposite end of the cradle, flank the ATP binding site (Fig. 4a). Site 2 comprises a cluster of three arginines (R137, R207, and R218), while site 3 consists of arginine R442 and lysine K444. Notably, none of the residues within these three sites were previously reported to be involved in the catalytic activity of PafA or Pup-binding. However, R207 is part of a loop that has been associated with facilitating PafA-substrate interaction previously[47]. The majority of residues within this loop, known as the α-loop, were believed to determine its orientation or participate in nucleotide binding and stabilization of the phospho-Pup ligation reaction intermediate. For R207, a direct link to substrate interaction was suggested,

because previous structural studies had shown a different orientation of R207 in PafA compared to the orientation of the equivalent arginine residue in the close structural homologue Dop. However, in the scope of that study, no interaction mechanism with a substrate was identified.

To investigate the role of the three identified clusters in substrate recognition, several alanine variants of PafA were generated. In vitro pupylation assays using PanB and FabD as substrates revealed significant pupylation defects for the site 2 triple alanine variant (R137A/R207A/R218A). Pupylation of FabD was not observed within the recorded time frame, and pupylation of PanB was reduced by 70% (Fig. 4b, red traces). The site 3 alanine double mutant showed a slight decrease in pupylation efficiency for both tested substrates, while no change in pupylation efficiency was observed for the site 1 alanine double mutant (Supplementary Fig. 6a).

To determine if these mutations specifically affect substrate recognition rather than Pup binding or activation, free lysine at high concentration was used, as it can serve as a substrate under these conditions[8]. Pupylation of free lysine at saturating conditions remained largely unaffected when comparing the PafA R207A variant and the full site 3 mutant with PafA, indicating that these residues do not play a direct role in the catalytic activity of PafA but rather influence the interaction with authentic substrates (Supplementary Fig. 6b).

We observed a slower reaction with the full site 2 PafA mutant, suggesting additional effects on the catalytic activity of the triple mutant. This observation may be attributed to the fact that residues in close proximity to site 2 interact with the nucleotide binding site[46]. The introduction of the three R to A mutations could potentially alter the orientation of these other critical residues and thereby impact activity. However, this alone would not be sufficient to explain the significant defects observed in substrate pupylation for the site 2 PafA mutant.

To further pinpoint the key residues within site 2, individual alanine and combinatorial alanine mutants were generated. All combinations that included R207A, as well as the single point mutation itself, resulted in a complete loss of FabD pupylation as exhibited by the triple mutant of site 2 (Fig. 4b, left). Replacing site 2 residues R137 and R218 with alanine individually resulted in a reduction in FabD pupylation rate, but complete pupylation could be

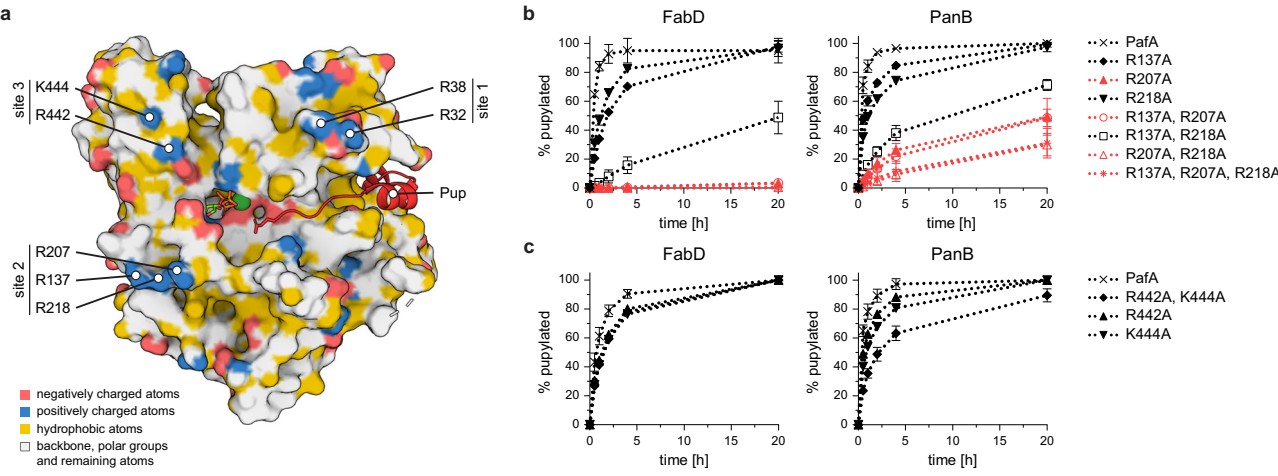

**Fig. 4 | Positively charged conserved residues bordering the PafA active site confer substrate binding. a** View of the active site of PafA (PDB: 4BJR) colored in the YRB scheme[68] with Pup (red, cartoon representation) bound to the Pup-binding groove and the C-terminal glutamate reaching into the active site. Highly conserved positively charged amino acids located around the rim of the active site are clustered into three surface patches referred to as sites 1–3 according to their location in the linear sequence. **b** and **c** Gel densitometric analysis of FabD (left panels) or

PanB (right panels) pupylation time courses using site 2 **b** or site 3 **c** PafA variants. Fraction of pupylated substrate is expressed in relation to the total amount of substrate present in the reaction. Pupylation experiments were carried out in independent triplicates, data are presented as mean values with error bars depicting standard deviation. Source Data (raw values from gel densitometric analysis and corresponding uncropped gels) are provided as a Source Data file.

**Table 1 | Apparent initial FabD pupylation rates of PafA and PafA variants**

| Substrate | PafA mutant | App. initial pupylation rate [h⁻¹] |
|---|---|---|
| FabD | PafA | $10.41 \pm 0.57$ |
| | R137A | $3.10 \pm 0.21$ |
| | R207A | No pupylation |
| | R218A | $5.05 \pm 0.15$ |
| | R137A, R207A | No pupylation |
| | R137A, R218A | $0.17 \pm 0.03$ |
| | R207A, R218A | No pupylation |
| | R137A, R207A, R218A | No pupylation |
| | R442A | $2.874 \pm 0.20$ |
| | K444A | $2.65 \pm 0.28$ |
| | R442A, K444A | $2.64 \pm 0.23$ |

Pupylation time courses recorded by SDS-PAGE and Coomassie-staining were quantified by gel densitometric analysis of the FabD and FabD-Pup bands. The apparent initial pupylation rate was determined by linear regression of the first four data points in dependence of the used PafA concentration. Fraction of pupylated substrate is expressed in relation to the total amount of substrate present in the reaction. Assays were carried out in independent triplicates. Representative gels are shown in Supplementary Fig. 6c.

achieved within 20 h, whereas the R137A/R218A double mutant was pupylated even more slowly and did not reach completion within this time frame. To assess this effect more quantitatively, apparent initial pupylation rates were determined by fitting the linear phase of the first 20 min of the pupylation reaction for those mutants showing measurable activity. Based on this analysis, both the individual mutants as well as the double mutant of R137A and R218A exhibited reduced apparent initial pupylation rates (Table 1, Supplementary Fig. 6c), with the double mutant displaying a 60-fold lower rate of $0.17\,h^{-1}$ (compared to $10.41\,h^{-1}$). Both residues appear to have an additive effect, since the individual mutants only reduced pupylation rates by half (R137A: $3.10\,h^{-1}$; R218A: $5.05\,h^{-1}$). This highlights the importance of the three residues comprising site 2, with R207 being the primary interactor, while the other two residues contribute to stabilizing the interaction, albeit to a lesser extent.

To validate these findings using another authentic substrate, we conducted pupylation assays with all variants using PanB. Similar to FabD, PanB pupylation was greatly diminished by the R207A mutation, albeit not to the same degree as the site 2 triple mutant. We observed a 50% reduction, consistent with previously reported results[47] (Fig. 4b, right). The double mutant R207A/R218A displayed a comparable reduction in pupylation efficiency as the triple site 2 mutant. Still, also in the case of PanB, R207 seemed to be the main interactor with a greater supporting role of R218 compared to pupylation of FabD. The single mutants R442A and K444A of site 3 exhibited the same level of reduction in pupylation for both FabD and PanB (Fig. 4c), but the combination of site 2 and 3 only resulted in further reduction of pupylation for PanB (Supplementary Fig. 6d).

### Mutation of PafA residues mediating substrate recognition in vitro affect in vivo pupylome formation and fitness under stress

To investigate the role of the identified residues in conferring substrate selectivity for a wide range of pupylation substrates, we assessed the equivalent mutations in ᴹˢᵐPafA in vivo (R130A, R193A and R204A corresponding to R137A, R207A and R218A of ᶜᵍˡPafA, respectively, Fig. 5a). A *Mycobacterium smegmatis* (Msm) strain lacking *pafA* was complemented with an integrative plasmid carrying either one of the ᴹˢᵐPafA mutant genes or wild type *pafA*. Western blot analysis of lysates using anti-Pup antibody was performed to detect all pupylated proteins present in the proteome. Strikingly, we observed a strong defect on pupylome formation when complementation was carried out with ᴹˢᵐPafA R193A (corresponding to R207A of ᶜᵍˡPafA), as hardly any pupylation was detected (Fig. 5b). The pupylome was also severely

reduced for the other two point-mutants, ᴹˢᵐPafA R130A and ᴹˢᵐPafA R204A, with the few remaining bands exhibiting a slightly different intensity pattern.

To investigate whether these mutations also impact the fitness of Msm under stress conditions, we exposed Δ*pafA* strains, complemented with either a *pafA* mutant or wild type *pafA*, to varying concentrations of the DNA damage-inducing agent mitomycin C (MMC). We previously showed that strains lacking parts of the PPS display reduced viability under MMC-induced stress[5]. Viability was assessed by monitoring the reduction of resazurin to resorufin through the oxidoreductase activity present in living bacterial cells. Our results demonstrated reduced viability for the Δ*pafA* strain as well as strains complemented with *pafA* point mutants R130A or R193A. However, the strain complemented with the *pafA* mutant R204A exhibited similar viability to the strain complemented with wild type *pafA* (Fig. 5c). Taken together, our observations show that residues within site 2 of PafA play a crucial role in the recognition of a diverse range of pupylation substrates in vivo.

### Swapping charges between PafA and FabD partially restores substrate recognition

To probe the putative charge-charge interactions between PafA and its substrate FabD, we performed residue swapping experiments between the two binding partners. In FabD, we mutated charged residues affecting pupylation (E147, D150, E176, and D177) in pairs to arginines, resulting in the variants R1 (FabD$_{E147R, D150R}$) and R2 (FabD$_{E176R, E177R}$). Due to its proximity to the pupylation site, E172 was excluded from these experiments. In PafA, we converted the putative interacting arginine residues to glutamates in site 2 (PafA$_{R137E, R207E, R218E}$). The charge-swapped PafA and FabD variants were then subjected to in vitro pupylation assays.

When using PafA to catalyze pupylation of FabD R1 or R2, we observed only minimal pupylation over a 20 h time course, while pupylation of FabD was completed within 4 h (Fig. 6, top). Furthermore, the observed minimal pupylation was likely due to off-site pupylation of the C-terminal Strep-tag used for affinity purification, rather than pupylation of the genuine target lysine. It has been noted previously that slow, unspecific pupylation can occur when PafA is not presented with a good target for pupylation[48]. In these cases, sterically unhindered lysines can diffuse into the active site of PafA and perform the nucleophilic attack on the phospho-Pup intermediate poised there to form the isopeptide bond. This unspecific reaction is outcompeted in the presence of a genuine substrate, and, therefore, Strep-tags do not interfere in pupylation assays with genuine substrates.

The site 2 mutant of PafA, in combination with FabD or FabD R1, exhibited significantly impaired pupylation (Fig. 6, bottom), similar to the observations made with the PafA mutants containing R207A (Fig. 4c). However, when using the E to R-swapped FabD R2 variant as a substrate for the R to E-swapped PafA site 2 variant, pupylation of target lysine K173 could be partially restored up to 50%. This provides supporting evidence for the postulated charge-charge interaction between site 2 arginine residues in PafA and E176 and D177 in FabD.

## Discussion

Pup ligase PafA acts as the sole ligase in the pupylation pathway of mycobacteria and other actinobacteria. Multiple studies have examined pupylated proteome data sets, identifying hundreds of potential pupylation substrates[21,23,24]. Attempts have been made to identify a substrate recognition pattern for PafA through bioinformatic analysis of the pupylome data sets. However, the results have been inconclusive[25–27]. In proteomics experiments for the identification of modified proteins, a common approach is to include an enrichment step during purification to separate the modified from unmodified proteins. However, this approach sacrifices information regarding the ratio between the product and the educt of the modification reaction.

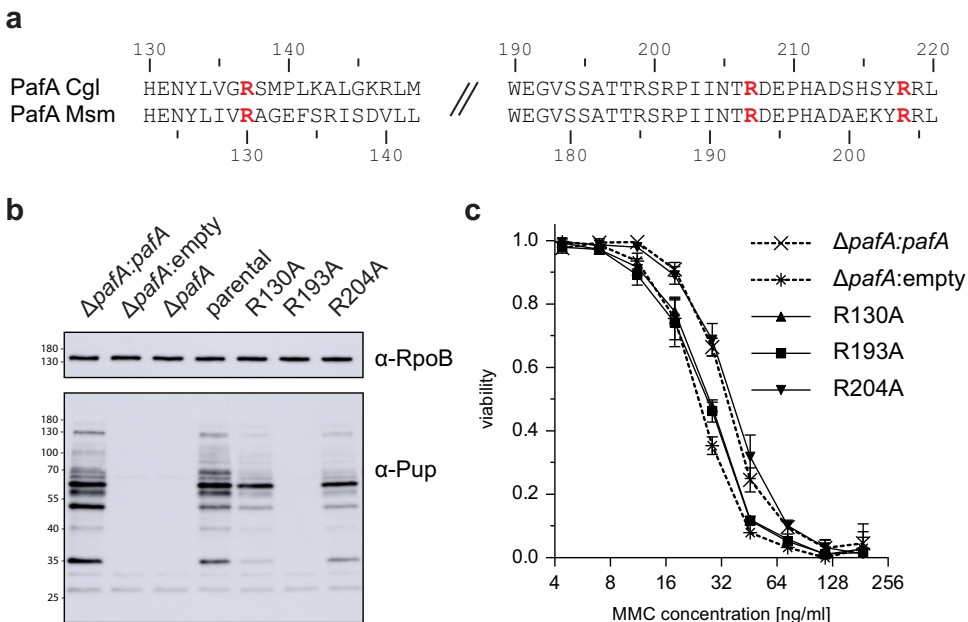

**Fig. 5 | PafA mutants of a conserved sequence patch bordering the active site show a global pupylation defect in vivo. a** Sequence alignment of site 2 region of [Cgl]PafA and *Mycobacterium smegmatis* (Msm) PafA. Mutated arginines are highlighted in red. **b** Immunoblot of Msm lysate using anti-Pup and anti-RpoB (loading control) antibodies. A Msm Δ*pafA* strain was complemented with either a mutant *pafA*, wild type *pafA* or an empty vector and grown to late exponential phase in liquid media before harvesting. In addition, the parental strain was grown and prepared alongside. Strains complemented with an alanine mutation at any of the site 2 residues (R130, R193 or R204) exhibited a strong defect in overall pupylation.

In case of complementation with *pafA* carrying the R193A mutation pupylation was completely abolished. Source Data (uncropped blots) are provided as a Source Data file. **c** Viability assay of complemented Msm Δ*pafA* under mitomycin C (MMC) induced DNA stress. The strains complemented with the *pafA* R130A and R193A mutants are sensitized to MMC to the same degree as the KO mutant Δ*pafA*. The R204A mutant, in contrast, shows no increased sensitivity. The assay was carried out in independent triplicates, data are presented as mean values with error bars depicting standard deviation. Source Data (absorbance reads) are provided as a Source Data file.

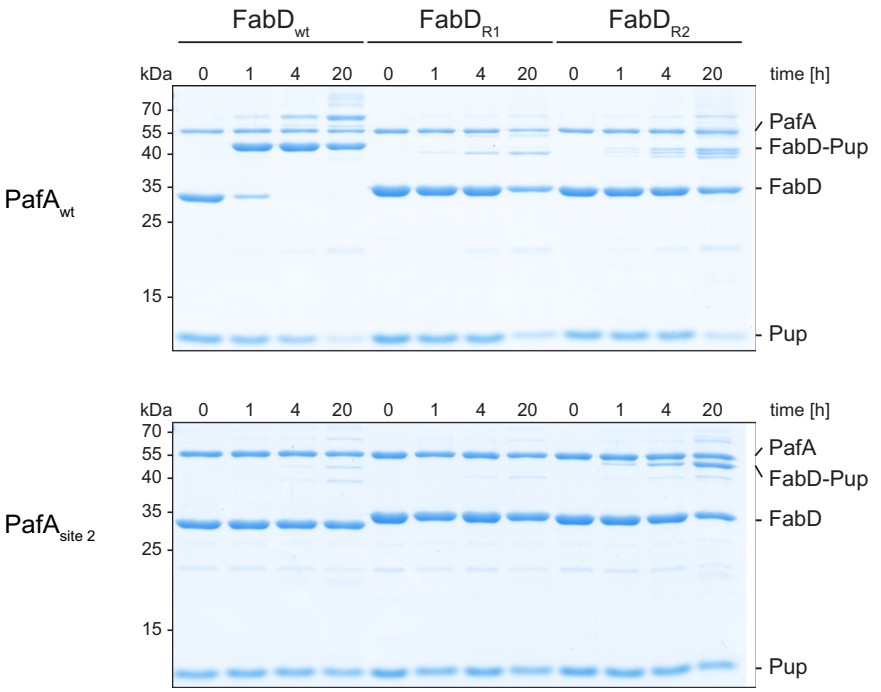

**Fig. 6 | Swapping complementary charges in the interaction site between PafA and FabD retains pupylation activity.** Top: Pupylation time courses catalyzed by PafA using different FabD variants (R1: E147R/D150R; R2: E176R/D177R) were monitored by SDS-PAGE and Coomassie-staining. FabD was readily pupylated within 4 h, while both charge-swapped mutants of FabD were only mildly pupylated within 20 h. As an unspecific side reaction, the Strep-tag on FabD was pupylated.

Bottom: Pupylation time courses catalyzed by a charge-swapped site 2 variant of PafA using different FabD variants. FabD and FabD variant R1 were only pupylated to a minor fraction, while pupylation of the R2 variant could be recovered up to 50% within the same time frame. As an unspecific side reaction, the Strep-tag on FabD was pupylated. Representative gels of three independent experiments are shown. Source Data (uncropped gels) are provided as a Source Data file.

Consequently, even proteins from populations modified only to a marginal degree can be identified as targets, even if their modification rate is well below a physiologically meaningful level. Such off-target pupylation reactions are expected in the physiological context of the cell. For instance, the well-studied authentic pupylation substrate PanB exhibits an apparent $K_m$ of ~14 μM for PafA, while free lysine has a three orders of magnitude higher $K_m$ of 22 mM[8]. Given that the concentrations of individual proteins within a proteome vary over at least four orders of magnitude, off-target effects on highly abundant proteins are to be expected[35]. Similarly, issues have been noted for the determined target proteomes of many other post-translational modifications (PTMs), particularly in the absence of explicit detection of the turnover stoichiometry[49]. It can therefore be argued that proteomics studies aiming to identify PTMs carry a not insignificant risk of producing false positives. Attempting to generate prediction models solely based on such data sets without carefully weighting the data set, will introduce a bias stemming from the inherent overrepresentation of highly abundant proteins, as exemplified by the pupylome data sets analyzed in this study. The identification of a common interaction motif by prediction alone without experimental validation is therefore not advisable.

The mentioned challenges associated with proteomic studies regarding PTMs are exacerbated in the case of pupylation by the physiological fate of the proteins as a consequence of pupylation. Proteasomal degradation continually eliminates pupylated proteins from the pupylome, making it difficult to identify good pupylation targets of low abundance. One such example is Lonely Guy (Log), a cytokinin riboside 5′-monophosphate phosphoribohydrolase, which was not detected in any of the pupylomes despite being a genuine and physiologically highly relevant pupylation target[31]. Another complicating factor arises from tryptic digestion, which typically generates the peptide fragments for mass spectrometry. This process can result in missed cleavage sites and consequently longer fragments when lysine residues are modified. In the case of PanB, pupylation-induced loss of the trypsin cleavage site at K212 generates a long and difficult to charge peptide that is not easily detected by mass spectrometry. Thus, it is not surprising that fragments containing pupylation sites of genuine substrates are often absent or that some pupylated proteins are not identified in pupylome studies. Additionally, pupylation, like many other PTMs, can be reversed by the deligase activity of Dop, adding another layer of complexity to the cellular steady-state pupylation level of any given target.

Due to these considerations, we opted to investigate the substrate specificity of PafA by examining the interaction of PafA with some of the well-known, authentic substrates in vitro. Our findings demonstrate that PafA exhibits defined selectivity for its substrates and does not indiscriminately ligate Pup to any accessible, surface-exposed lysine under physiological conditions. One way to achieve selectivity would be that PafA recognizes a short sequence degron presented by its substrates. Many E3 ligases recognize such degrons which frequently do not include the target lysine but are located distal to it[50]. In such cases, the ligases usually employ a recognition domain that is connected to the catalytic domain. However, no such motif could be identified in any of the members of the pupylome. Furthermore, our analysis revealed that PafA requires only a limited subset of conserved residues flanking the active site in order to recognize its substrates. This suggests that PafA is also unlikely to recognize substrates based on features located far away from the targeted lysine. Instead, the positively charged residues we identified guide substrates into the right orientation and engage into electrostatic interactions with negatively charged residues located on the protein surface surrounding the target lysine. For these reasons, we suggest that PafA in most cases recognizes its substrates in a degron-independent manner. This is also supported by the observation that Msm mutant strains carrying *pafA* point mutations at the positions of the identified positively charged residues bordering the active site, diminish the entire pupylome rather than change its composition. However, we cannot exclude some exceptions to this rule, where certain substrates are recognized in presence of an additional factor that serves as an ancillary factor for the substrate. For instance, a substrate might be recognized only in complex with another protein, or a protein complex might require disassembly to reveal the pupylation site.

For substrate recognition, PafA relies on specific arginine residues at site 2, particularly R207, which flank the nucleotide-binding end of the β-sheet cradle opposite the C-terminal domain of PafA. In the absence of R207, in vitro pupylation of FabD and PanB is strongly impaired, and in vivo pupylome formation cannot be detected. However, the activated phospho-Pup species can still be formed, as demonstrated by lysine conjugation at high concentration. This demonstrates that residue R207 is essential for recognition of most substrates. Mutation of the other two residues within site 2, R137 and R218, lead to a reduced pupylation rate of FabD and PanB as well as a diminished pupylome. An accessory role is played by site 3 that is located in strand β7 of the C-terminal domain of PafA also flanking the nucleotide binding site. For some substrates, like PanB, site 3 plays a bigger role than it does for example for FabD. Hence, we propose that in addition to R207, required for the recognition of all substrates, different substrates rely on distinct sets of positively charged residues within site 2 and 3 to interact with PafA. Furthermore, it remains possible that additional residues not identified in this study also support binding of certain substrates.

While PafA and Dop are close structural homologues, by virtue of their opposing roles they do not require the same mode of substrate recognition. Dop likely recognizes substrates through the Pup-tag and binds Pup more tightly than does PafA[46]. R207 (R227 in Dop) is conserved between the two and is essential for Dop activity. It was shown to take part in stabilizing the phospho-Pup intermediate during the depupylation reaction[10]. In contrast, the PafA R207A variant is able to form the phospho-Pup species, indicating R207 does not have the same role in PafA as in Dop. Interestingly, a study converting Dop into a Pup ligase demonstrated that the function of R227 in Dop can be altered to substrate recognition by mutation of nine residues preceding R227[51].

On the example of FabD we could show that a limited set of surface-exposed charged residues surrounding the pupylation site is sufficient to make this protein a good substrate for PafA. Through electrostatic interactions, FabD transiently associates with PafA site 2, allowing the nucleophilic attack of the ε-amino group of the lysine side chain to occur. After the formation of the isopeptide bond, the pupylated FabD product dissociates from the active site, enabling the initiation of a new pupylation cycle.

The observation that a few or even a single residue can convert good substrates into exceedingly poor ones and vice versa agrees well with the previously measured $K_m$ of PafA for a substrate of around ~14 μM. Assuming rapid dissociation of the enzyme substrate complex and a slow turnover rate, the $K_m$ serves as a good estimate for the $K_d$ of the enzyme-substrate interaction. This suggests that the binding energy of PafA binding to PanB is approximately −26 to −30 kJ/mol[8], indicating the release of energy in the range of one to a few hydrogen bonds for enzyme-substrate complex formation. This is in line with our finding that only a few well-positioned residues are essential for efficient pupylation.

Small interfaces provide evolution with greater flexibility and more readily lead to alternate uses and novel targets that can be controlled with pupylation. Indeed, we already noted previously that the pupylation machinery within different actinobacteria exhibits signatures of rapid divergent evolution[52]. In absence of structural data of a PafA-substrate complex, such a small footprint encoded in the tertiary structure will make it difficult to construct a robust pupylation prediction algorithm. However, the small footprint also makes for the

exciting opportunity, that a change of the interaction surface introduced by another PTM, for example phosphorylation, can cause enough of a change to alter substrate specificity of PafA. Subsequently this could modulate turnover of proteins and thus allow Actinobacteria to adapt to the ever-changing environments they are exposed to.

## Methods

### Data analysis

For the statistical analysis of the pupylome data, Mtb proteins are considered "pupylated" if they are reported in any of the pupylome data sets within the pupylated fraction (irrespective of whether a target lysine was identified or not)[21,23,24] otherwise they are considered "not pupylated". As Mtb protein abundance estimates we used protein abundance during exponential growth determined elsewere[35]. The authors report abundance as copies/cell. We transformed these numbers back into a cytosolic protein concentration ($\mu$mol/L) for a more intuitive interpretation in the context of enzyme kinetics using the stated cell volume of $0.5 \mu m^3$ for Mtb[53]. For molecular weight the reported UniProt mass in kDa was used, protein sequences were extracted from the *Mycobacterium tuberculosis* H37Rv reference proteome[54]. Two sample Kolmogorov-Smirnov test and Welch's *t*-test for the difference in mean protein abundance, mean molecular weight, mean total lysine count and mean lysine density for the pupylated and non-pupylated proteome was calculated with the Python implementation in the scipy.stats library (version 1.7.3).

The solvent accessible surface area for the lysine residues in a given protein structure context was estimated with the dot density technique as described[55,56]. The Python implementation in the MDTraj library (version 1.9.3) was applied to the following PDB structures: 2QC3 (FabD)[44], 7PXC (Mpa)[12], 1OY0 (PanB)[57], 4R43 (PckA)[58], 1F8I (IdeR)[59], 1GR0 (Ino1)[60], 1FX7 (Icl1)[39], 4PSK (RecA)[61], AF-O05306-F1 [https://alphafold.ebi.ac.uk/entry/O05306] (Log)[40,41], AF-P9WIS5-F1 [https://alphafold.ebi.ac.uk/entry/P9WIS4] (KGD)[40,41], 2CDN (Adk)[62], 5E0S (ClpP2)[63], 5KVU (Icd2), 4TVO (MDH)[64], 4BJR (PafA)[45], 1UBQ (Ub)[65].

In vitro pupylation of selected substrates was assessed from in vitro pupylation assays after approximately 4 h. We also included known in vitro pupylation targets widely used in the literature or substrates where accumulation in PPS member knockout studies was observed (see Supplementary Table 1). Structure visualization and distance measurements were carried out in PyMol (version 2.4.1).

### Molecular cloning and knockout strain generation

To prevent self-pupylation of [Cgl]PafA on lysine residues K106, K172, and K302, which were identified as pupylation targets through mass spectrometry, the residues were mutated to arginine by using the QuikChange Lightning Multi Site-Directed Mutagenesis kit (Agilent). [Cgl]PafA[K106R, K172R, K302R] was used in all in vitro experiments and for better readability labeled as PafA, since no apparent effect on substrate pupylation was observed (Supplementary Fig. 5a).

Mutants of [Cgl]PafA and [Mtb]FabD3KR were generated by Q5 Site-Directed Mutagenesis (NEB) followed by a DpnI digestion and T4 ligation step (KLD reaction, NEB) using the parental plasmids as template. For in vivo experiments, the corresponding residues in [Msm]PafA were mutated by Q5 Site-Directed Mutagenesis followed by KLD reaction using a pFLAG_attP-derived plasmid for site-specific integration in Msm carrying the *pafA* gene. The empty pFLAG_attP vector was a gift from Markus Seeger. Plasmids were validated by sequencing using commercially available primers for the T7 promotor (Microsynth) or LK27_fw (see Supplementary Table 2) for pFLAG_attP based constructs.

To generate a markerless deletion of *pafA* ($\Delta pafA$), the suicide plasmid pGOAL19 was generated containing the 1500 bp up- and downstream chromosomal region of *pafA*, as well as the first and last three amino acids of PafA. The PCR products (for primer sequences see Supplementary Table 2) were designed with an overhang for ligation with plasmid cut by XmnI and were ligated using Gibson assembly (NEBuilder HiFi DNA Assembly Master Mix, NEB). NEB5$\alpha$ cells were transformed with the Gibson assembly reaction mix, plated on LB containing hygromycin (100 $\mu$g/mL), and successful transformants were sequenced (Microsynth). Msm SMR5 cells were transformed with the suicide plasmid for allelic exchange mutagenesis according to Parish and Roberts[66]. Briefly, 200 $\mu$L of competent Msm SMR5 cells were transformed with 1.5 $\mu$g suicide plasmid by electroporation (2.5 kV). Electroporated cells were immediately recovered in 5 mL shaking cultures of 7H9 complemented with 0.2% (v/v) glycerol and 0.05% (v/v) Tween-80 for 4 h at 37 °C, after which cells were plated on 7H10 plates supplemented with 0.5% (v/v) glycerol and hygromycin (50 $\mu$g/mL). Following 3 days of incubation at 37 °C, single-crossover (SCO) colonies were identified by 5-brom-4-chlor-3-indoxyl-$\beta$-D-galactopyranosid (X-gal) underlay (200 $\mu$L of 0.4% (w/v) X-gal in DMSO were spread underneath the agar). SCOs turn blue after an overnight incubation at 37 °C due to the expression of $\beta$-galactosidase encoded on the suicide plasmid. SCOs were picked and grown in 5 mL 7H9 supplemented with 0.2% (v/v) glycerol and 0.05% (v/v) Tween-80 as well as hygromycin (50 $\mu$g/mL) to an $OD_{600}$ of 0.7. Then, the cells were plated on 7H10 plates supplemented with 0.5% (v/v) glycerol and 2% (w/v) sucrose in 1:10 and 1:100 dilutions. Expression of the *Bacillus subtilis* levansucrase SacB, encoded on the suicide plasmid, is lethal in the presence of sucrose. After incubation at 37 °C for 3 days the X-gal underlay was repeated as described as a second round of selection. Double crossover (DCO) cells should stay white, as the suicide plasmid should be lost during homologous recombination. DCO cells were screened through colony PCR as well as sequencing of a PCR product generated with primers annealing 2500 bp up- and downstream of *pafA*. In addition, deletion of *pafA* was verified through RT-PCR.

Sequences of used primers are listed in Supplementary Table 2.

### Protein expression and purification

[Cgl]PafA3KR-H$_6$, [Mtb]H$_{10}$-Trx-TEV-Pup, [Mtb]PanB-Strep and [Mtb]FabD3KR-Strep were expressed and purified as described[43,46,67]. Protein sequences can be found in Supplementary Table 3. Proteins were expressed from IPTG-inducible plasmids in *E. coli* Rosetta (DE3) cells (Invitrogen). Bacteria were grown for 4 h at 37 °C, induced with 1 mM IPTG and further incubated over night at 23 °C. Cells were lysed in buffer P (50 mM HEPES-KOH, pH 7.5, 150 mM NaCl, 1 mM EDTA. 1 mM DTT, 10% (v/v) glycerol) containing 1 mM PMSF (Roche) as well as complete protease-inhibitor cocktail (Roche) with high pressure shear force using a Microfluidizer M110-L device (Microfluidics; 5 passes, 11,000 psi chamber pressure). Lysate was cleared by centrifugation for 60 min at 47810 × *g* and 4 °C. Pup-containing lysate was heated to 80 °C for 10 min, precipitates were removed by centrifugation for 20 min at 14680 × *g*. His-tagged proteins were subjected to Ni$^{2+}$-affinity chromatography using a 5 mL, manually packed IMAC Sepharose 6 FF column (Cytiva). Impurities were removed with buffer P containing 50 mM imidazole and eluted with buffer P containing 250 mM imidazole. Strep-tagged proteins were subjected to a 5 mL manually packed Strep-Tactin XT Superflow column (IBA Lifesciences) and eluted in buffer P containing 50 mM biotin. PafA-, FabD- or PanB-containing fractions were pooled and dialyzed against buffer P overnight at 4 °C. Pup-containing fractions were pooled, supplemented with His-tagged TEV protease and dialyzed overnight against buffer P at 4 °C. The tag together with the protease was removed by Ni$^{2+}$ affinity chromatography. PafA was further purified by size exclusion chromatography using a Superdex 200 gel filtration column in 50 mM HEPES-KOH, pH 7.5, 150 mM NaCl, and 10% (v/v) glycerol. Pup was further purified by size exclusion chromatography using a Superdex 75 gel filtration column (Cytiva) in 50 mM HEPES-KOH, pH 7.5, 150 mM NaCl, and 1 mM EDTA. PanB was further purified by size exclusion chromatography using a Superose 6 gel filtration column (Cytiva) in 50 mM HEPES-KOH, pH 7.5, 150 mM NaCl, and 10% (v/v) glycerol. FabD was further purified

by size exclusion chromatography using a Superdex 75 gel filtration column (Cytiva) in 50 mM HEPES-KOH, pH 7.5, 150 mM NaCl, 1 mM EDTA, and 10% (v/v) glycerol. All proteins where stored at −20 °C until further use. Protein concentrations were determined by absorbance spectroscopy at 280 nm. Generated variants were purified according to the same protocol as the wild type. $^{Bov}$Ubiquitin was obtained commercially (Sigma-Aldrich, #U6253).

### Substrate pupylation assays

1 µM $^{Cgl}$PafA was mixed with 12 µM $^{Mtb}$Pup and 6 µM substrate (or 80 mM L-lysine) and adjusted to 50 µL total reaction volume with pupylation buffer (50 mM HEPES-KOH, pH 8.0, 150 mM NaCl, 20 mM MgCl$_2$, 1 mM DTT, 10% (v/v) glycerol). The reaction was started by the addition of 5 mM ATP and incubated at 30 °C in a thermocycler with heated top to prevent condensation. 5 µL aliquots were taken at the indicated time points and the reaction was quenched by the addition of 6× SDS loading dye. The formation of pupylated substrates was monitored by SDS-PAGE followed by Coomassie-staining and subsequently analyzed densitometrically using the GelAnalyzer software (version 19.1). The fraction of pupylated substrate is expressed in relation to the total amount of substrate present in the reaction. Assays were carried out at least in independent triplicates.

### Protein S-carbamidomethylation and CD spectroscopy

FabD3KR-Strep or PanB-Strep respectively were dialyzed into the degassed buffer (200 mM Tris-HCl, pH 8.5, 6 M Gdm-Cl, 10 mM Na$_2$EDTA, and 1 mM DTT), and 17 mM iodoacetamide in 200 mM (NH$_4$)CO$_3$ was added. The reaction was carried out in the dark at room temperature for 30 min and stopped by exchanging the buffer to 200 mM Na$_2$PO$_4$, pH 8.0 using a PD-10 desalting column (Cytiva). For evaluation of the secondary structure of the S-carbamidomethylated proteins, samples were diluted to 0.2 mg/mL and measured on a Jasco J-710 spectropolarimeter in a Hellma QS Macro Cell with 1 mm path length. Three consecutive wavelength scans from 205 to 260 nm were taken and mean residue ellipticity $\theta_{MRW}$ was calculated according to the following equation.

$$\theta_{MRW} = \frac{\theta \cdot 100 \cdot m}{c \cdot d \cdot n}$$

($\theta$: measured ellipticity, $m$: molecular weight (in Da), $c$: protein concentration, $d$: thickness of cuvette (in cm), $n$: number of amino acids)

### In vivo assessment of pupylation defects

An electrocompetent Msm $\Delta pafA$ strain was transformed with integrative plasmids carrying $pafA$ with the respective point mutations together with a pMyNT derived vector carrying the L5 integrase for site-specific integration of $pafA_{mut}$. Complemented strains were grown in 5 mL shaking cultures at 37 °C in 7H9 medium complemented with 0.2% (v/v) glycerol and 0.05% (v/v) Tween-80 containing Apramycin (50 µg/mL). Samples were taken at OD$_{600}$ 1.0, spun down and cell pellets were frozen for further usage. Cell pellets were lysed by bead beating in buffer L (20 mM HEPES-KOH, pH 7.0, 150 mM KCl, 2 mM EDTA) using 500 mg zirconium oxide beads (0.15 mm). Beating was performed on a Bertin Instruments MiniLys bead beater (3 × 30 s, 6 m/s with a cool-down phase of 30 s on ice in between). Insoluble parts were removed by centrifugation and total protein concentration was determined using Bradford assay. In total 10 µg of total protein was run on 4–20% PROTEAN stain-free PAGE gels (BioRad) and transferred to Immobilon-P PVDF membrane (Merck). Immunoblot analysis was carried out using anti-Pup antiserum (1:10,000)[7]. For detection, goat anti-rabbit IgG antibody conjugated to horseradish peroxidase (HRP) (abcam #ab6721) was used together with BioRad Clarity Western ECL Substrate (1:1 ratio; BioRad) as a substrate for the HRP. To control for

equal loading, immunoblotting against RNA polymerase subunit β (RpoB) was performed using anti-RpoB antibody (1:5,000, BioLegend, clone 8RB13). Detection was carried out using HRP-conjugated anti-mouse IgG antibody (1:5,000, Promega #W4021).

The resazurin viability assay was performed as previously described[5]. In short, complemented Msm $\Delta pafA$ strains were grown to an OD$_{600}$ of 0.6 in 7H9 medium. Cells were diluted into fresh medium containing the indicated concentration of MMC and incubated at 37 °C for 24 h. Resazurin was added to a final concentration of 20 µg/mL, and incubation was continued at room temperature for 72 h. Absorbance at 605 nm was measured and values were normalized within each dataset.

### Reporting summary

Further information on research design is available in the Nature Portfolio Reporting Summary linked to this article.

## Data availability

The following protein structures from the Protein Data Bank with PDB accession codes 7PXC, 1OY0, 2QC3, 4R43, 1FX7, 1GR0, 1F8I, AF-O05306-F1 [https://alphafold.ebi.ac.uk/entry/O05306], 4PSK, AF-P9WIS5-F1 [https://alphafold.ebi.ac.uk/entry/P9WIS4], 2CDN, 5E0S, 5KVU, 4TVO, and 4BJR were used throughout this study. Source data are provided in this paper.

## Code availability

The script used for evaluation of the pupylomes[21,23,24] as well as the protein abundance data set[35] can be found here: https://www.github.com/EWBlab/PupStatistics; https://doi.org/10.5281/zenodo.8214992.

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

## Acknowledgements
We thank members of the Weber-Ban group for critically reading the manuscript. We are also grateful for the support of the Functional Genomics Center Zurich with acquisition and evaluation of mass spectrometry data and to Markus Seeger for gifting of the pFLAG_attP vector. This work was supported by Swiss National Science Foundation grants 31003A_163314 and 310030_185250 awarded to E.W.B.

## Author contributions
M.F.B., C.L.D., and E.W.B. conceived and planned the study. M.F.B. and C.L.D. performed the cloning work for this study. C.L.D. performed the bioinformatic data analysis. M.F.B. purified all proteins and performed all in vitro and in vivo experiments. L.M.L.K. created the Δ*pafA* strain and assisted with in vivo experimentation. T.T.S. helped with in vitro experimentation. M.F.B., C.L.D., and E.W.B. interpreted the results, analyzed the data, and wrote the manuscript.

## Competing interests
The authors declare no competing interests.
