## [Peer Review File · Nature Communications]

REVIEWER COMMENTS

Reviewer #1 (Remarks to the Author):

The authors present a new structural interpretation of the specificity and selectivity of pupylation, the bacterial parallel mechanism of ubiquitination in eukaryotes. Overall, I find the question really interesting; understanding various ways regulated protein degradation is actuated is extremely valuable in the basic science sense as well as for biotechnology. I find the presented data convincing that the central pupylation enzyme, PafA, utilizes electrostatic interactions to recognize certain substrates in order to pupylate them. I have a few points that I think might be relevant to interpreting the results and some additional suggestions to address these.

Major points:

- Ubiquitination in eukaryotes relies mostly on two aspects of the substrate for proper recognition by E3 ligases (at least in the majority of cases): an available lysine in the substrate protein where Ub gets attached and a degron, i.e., a short linear sequence motif in the substrate which is specifically bound by a substrate/target recognition module of the E3 ligase. The ubiquitination site (the lysine) usually does not show very strong sequence patterns, similarly to the pupylation-target lysines explored in the presented study, as the specificity comes from the degron, which might be very far away from the lysine. Throughout the work, the authors implicitly assume that pupylation degrons do not exist, but is there any evidence for this? While the direct electrostatic interaction-mediated binding between PafA and some substrates is convincing, could other substrates be recognized in a degron-dependent way? This is actually an alternative hypothesis that can explain several of the findings, e.g., why the substrate “alignment [around the lysine] shows no obvious recognition motif that would be encoded in the primary structure” or why ubiquitin cannot be pupylated.
- Is it possible that at least for some substrates, the interaction with PafA is mediated by a third ancillary molecule? If this is the case for a substrate, then the in vivo pupylation might happen while the in vitro one might not since the third necessary molecule isn't present. The authors would categorize such substrates as ‘bad’, implying that this is a false positive of the large-scale pupylation detection. Unless there is strong reason to believe that this is not feasible and all substrates bind directly to PafA, it might be better to rework the good/intermediate/bad classification
- In my opinion, the structural features the authors identify are indeed part of the recognition for the selected substrates. However, this binding mode seems to be true for probably only a subset of substrates, as some high confidence pupylated lysines have very high accessible surfaces, close to the theoretical maximum in a GGXGG sequence context (figure 2a). In general, it would be interesting to separate known target lysines according to whether they are in a structured context or part of a disordered protein region. This is relatively easy to do with either a standard disorder prediction tool or based on AlphaFold structures (e.g., see <https://www.nature.com/articles/s41594-022-00849-w>). For the disordered pupylated lysines, what would provide specificity in the PafA interaction?

- The authors present compelling indirect evidence that charged residues are key for certain PafA:substrate interactions. However, it would be nice to see some direct structural assessment of this. I understand that the determination of the complex structure might not be feasible, but it should be possible to build a structural model of the complex using some docking method or AlphaFoldMultimer, especially so given that both partners are stable, well-folded proteins (disordered protein docking is much more difficult).

Minor points:

- many sentences from the first results section would fit better in the introduction, especially the ones with references

- on figure 1b, the authors assess the possible influence of molecular weight on pupylation state and do not find a significant connection. First, it might be better to use the Kolmogorov-Smirnov test to test the difference between the two distributions. Second, since the rationale behind looking at molecular weight is that “larger proteins harbor more lysine residues on average”, why not use the number of surface accessible lysines directly?

- in the results, the authors state that they modified the substrates by S-carboxyamidomethylation, but in figure S1c, the modification is defined as alkylation. While this is a really clever indirect way of testing for the role of structural integrity, having a degron motif would be an alternative interpretation if the modified residue happens to be inside the degron, which would also impair binding, albeit not for structural reasons.

- some alanine mutations around the pupylated lysine severely decrease pupylation efficiency. Did the authors check (e.g., with CD, which they use in other parts of the paper) these proteins to make sure the protein has the same fold?

- the authors identify key charged residues important for the interaction; however, based on the structures, several hydrophobic residues are close to the target lysine as well, and the importance of possible hydrophobic interactions in the PafA:substrate binding would not necessarily show up in an alanine scanning experiment. Maybe the authors could mention that hydrophobics might also play important roles in the binding event (which could be more directly assessed using e.g., FoldX on a model complex structure and checking whether the experiments and computationally calculated $\Delta\Delta G$ values are in line). This might also provide the basis for claims that the authors already make, e.g., “only a few well-positioned residues are essential for efficient pupylation”.

- what is SMR5 in figure 5b?

- I feel that some claims in the discussion are a bit too broad and might benefit from more careful wording. E.g. “PafA mainly utilizes site 2 arginine residues for substrate recognition, in particular R207...”

- this is probably only true for some substrates, while there might be others that bind with a different binding mode (especially if they are disordered).”

This is not something to correct, I just wanted to say that I find the point that the authors make at the end of the discussion to be really interesting: “the small footprint also makes for the exciting opportunity, that a change of the interaction surface introduced by another PTM, for example phosphorylation, can cause enough of a change to alter substrate specificity of PafA”. This indeed sounds like an exciting prospect and I hope the authors can explore this in the future.

Reviewer #2 (Remarks to the Author):

The manuscript “Electrostatic interactions guide substrate recognition of the prokaryotic ubiquitin-like protein ligase PafA” by Block et al is a very interesting and important study, because there is a general lack of understanding how these modified protein substrates, which are thereby marked for degradation, are recognized and selected for pupylation.

To investigate the molecular basis of pupylation by PafA the authors revisited and analyzed massspec studies on the pupylome, identified and classified substrate proteins in combination with in vitro pupylation experiments (Fig 1&2). With further biochemical and bioinformatic analysis with structure models (also using alpha-fold structural models) they determined that the substrate recognition mechanism is based on the surface located accessible specific Lysine residue in combination with charged patches on the surrounding surface area using structure function analysis with protein variants both on the substrate FabD (Fig 3) and the PafA side (Fig 4). Similar PafA variants had a clear in vivo phenotype displaying lowered to no pupylation and a defect in DNA repair comparable to a Δ pafA phenotype (Fig 5). Finally in one experiment they could switch the charge on one of the cognate patches on PafA and FabF reconstituting the pupylation activity of the PafA(site2) with the FabD(R2) substrate (Fig 6).

The identified cognate patches on PafA were not all equally important for the recognition of different tested substrate proteins, which might be the basis for different recognition patterns, mediating the recognition and substrate selection.

Comments and remarks

-The pupylation of substrate proteins is probably mediated by the activities of the modifying PafA and the de-modifying Dop Enzyme.

Since PafA and Dop are homologs, an apparent question, which could be discussed, is whether the substrate recognition mechanism of Dop could be analogous to the described substrate interaction and recognition mechanism of PafA with substrates like FabD.

-Minor comments

“scatter” should be “circle”?

-p 6 l151 “larger scatters” (Fig 1c) The term “scatter” should be “circle”?

-Fig 1 p 31 l 836 “scatter size” should be “circle size” or “size of the circle”?

-p8 l180 & 181 “filled scatter” should be “filled circle”, “open scatter” should be “open circle”?

- p7 l 155-6 “circle” instead of “scatters”?

-Fig 1 PanB is shown in Fig 1a and 1c. Where are the in vitro substrates Adk, Mdh or ClpP2 (depicted in Fig1a) located in Fig 1c?

-Fig 1 Fig legend p 31 l 827 the reference of the mass spectrometry study is not “1”

-Fig 4 site 3 aa residues on PafA are depicted in 4A but no data is shown in 4b or c. Maybe somehow depict the data shown in suppl Fig 4a in Fig b or c?

-Text p7 l 155-6 “In contrast, there is a strong correlation between reactivity and steady-state accumulation in vivo (lime green scatters) in pupylation-deficient strains...”

Is the “reactivity” actually “in vitro reactivity”?

-Text p7 l 158-161 These two sentences (“taken together.... pupylated by PafA”) appear to be redundant both with the text before and with the following text starting the new paragraph. Maybe one could just delete these two sentences here?

Reviewer #1 (Remarks to the Author):

The authors present a new structural interpretation of the specificity and selectivity of pupylation, the bacterial parallel mechanism of ubiquitination in eukaryotes. Overall, I find the question really interesting; understanding various ways regulated protein degradation is actuated is extremely valuable in the basic science sense as well as for biotechnology. I find the presented data convincing that the central pupylation enzyme, PafA, utilizes electrostatic interactions to recognize certain substrates in order to pupylate them. I have a few points that I think might be relevant to interpreting the results and some additional suggestions to address these.

Major points:

C1: Ubiquitination in eukaryotes relies mostly on two aspects of the substrate for proper recognition by E3 ligases (at least in the majority of cases): an available lysine in the substrate protein where Ub gets attached and a degron, i.e., a short linear sequence motif in the substrate which is specifically bound by a substrate/target recognition module of the E3 ligase. The ubiquitination site (the lysine) usually does not show very strong sequence patterns, similarly to the pupylation-target lysines explored in the presented study, as the specificity comes from the degron, which might be very far away from the lysine. Throughout the work, the authors implicitly assume that pupylation degrons do not exist, but is there any evidence for this? While the direct electrostatic interaction-mediated binding between PafA and some substrates is convincing, could other substrates be recognized in a degron-dependent way? This is actually an alternative hypothesis that can explain several of the findings, e.g., why the substrate “alignment [around the lysine] shows no obvious recognition motif that would be encoded in the primary structure” or why ubiquitin cannot be pupylated.

A1: The reviewer brings up a very valid point. Many E3 ligases recognize a short sequence (degron) that does not include the target lysine and employ a recognition domain that is connected to the catalytic domain. The targeted lysine is then determined by structural constraints imposed by the linkage

between these two domains. However, we believe that a similar mechanism in PafA is less likely. PafA as a small globular protein is not big enough to probe sequences distant from the actual pupylation site. Additionally, amino acid conservation is limited to the immediate surrounding of the active site of PafA, specifically the rim around the active site and the Pup binding groove, indicating that these areas confer the function of the ligase. For these reasons, we believe PafA in most cases recognizes its substrates in a degron-independent manner. As we demonstrated in this manuscript in Figure 5b, substrate recognition through the surrounding of the active site (site 2) seems to be the rule rather than the exception. However, we cannot exclude exceptions to this rule, for example in conjunction with dedicated, rather small adaptor domains for recruitment to the ligase.

To address these points, we added a short paragraph in the discussion. “One way to achieve selectivity would be that PafA recognizes a short sequence degron presented by its substrates. Many E3 ligases recognize such degrons which frequently do not include the target lysine but are located distal to it. In such cases, the ligases usually employ a recognition domain that is connected to the catalytic domain. However, no such motif could be identified in any of the members of the pupylome. Furthermore, our analysis revealed that PafA requires only a limited subset of conserved residues flanking the active site in order to recognize its substrates. This suggests that PafA is also unlikely to recognize substrates based on features located far away from the targeted lysine. Instead, the positively charged residues we identified guide substrates into the right orientation and engage into electrostatic interactions with negatively charged residues located on the protein surface surrounding the target lysine. For these reasons, we suggest that PafA in most cases recognizes its substrates in a degron-independent manner. This is also supported by the observation that Msm mutant strains carrying *pafA* point mutations at the positions of the identified positively charged residues bordering the active site, diminish the entire pupylome rather than change its composition. However, we cannot exclude some exceptions to this rule, where certain substrates are recognized in presence of an additional factor that serves as an ancillary factor for the substrate. For instance, a substrate might be recognized only in complex with another protein, or a protein complex might require disassembly to reveal the pupylation site.”

C2: Is it possible that at least for some substrates, the interaction with PafA is mediated by a third ancillary molecule? If this is the case for a substrate, then the *in vivo* pupylation might happen while the *in vitro* one might not since the third necessary molecule isn't present. The authors would categorize such substrates as 'bad', implying that this is a false positive of the large-scale pupylation detection. Unless there is strong reason to believe that this is not feasible and all substrates bind directly to PafA, it might be better to rework the good/intermediate/bad classification

A2: The participation of an additional required factor for pupylation of a large subset of substrates seems unlikely as several studies could show pupylation of natural (and unnatural) substrates *in vitro* as well as *in vivo* when only PafA and Pup were supplied (for instance reconstituted in *E. coli* (Cerdeira-Maira et al., “Reconstitution of the Mycobacterium tuberculosis pupylation pathway in *Escherichia coli*” EMBO Rep, 2011) or proximity labeling in mammalian cell lines (Liu et al., “A proximity-tagging system to identify membrane protein-protein interactions” Nat Methods, 2018)). In fact, those substrates for which either steady-state accumulation could be demonstrated in a proteasome knockout or for which a physiological role of pupylation could be demonstrated, are usually very efficiently pupylated *in vitro* by just PafA alone. However, this does not exclude the existence of certain natural substrates which might only be recognized in presence of an additional factor that then serves as an ancillary factor for the substrate rather than for PafA. For instance, it could be recognized only in complex with another

protein, or a protein complex could require disassembly to reveal the pupylation site. Indeed, in such cases the classification of good/intermediate/bad might change. We are now mentioning this possibility in our discussion.

C3: In my opinion, the structural features the authors identify are indeed part of the recognition for the selected substrates. However, this binding mode seems to be true for probably only a subset of substrates, as some high confidence pupylated lysines have very high accessible surfaces, close to the theoretical maximum in a GGXGG sequence context (figure 2a). In general, it would be interesting to separate known target lysines according to whether they are in a structured context or part of a disordered protein region. This is relatively easy to do with either a standard disorder prediction tool or based on AlphaFold structures (e.g., see <https://www.nature.com/articles/s41594-022-00849-w>). For the disordered pupylated lysines, what would provide specificity in the PafA interaction?

A3: It is correct that some of the pupylated lysine residues show a high accessibility. We carried out the suggested analysis and found that the pupylated lysines are mostly part of alpha helices or short connecting loops. We used IUPred short to predict the disorder of all proteins shown in Fig. 2a except for Ub. Most of the shown lysine residues, pupylation target or not, are located within ordered regions. The only exceptions are IdeR K229 with an IUPred score of 0.86 and Icd2 K112 and K653 with IUPred scores of 0.52 and 0.56 respectively. K229 in IdeR is the penultimate C-terminal residue but shown in solved crystal structures as well as in the AlphaFold prediction (pLDDT > 90) to be structured. K112 and K653 of Icd2 are in structured regions based on the determined crystal structure as well as the AlphaFold prediction (pLDDT > 90). The mean score for all analyzed lysines is 0.23 and therefore well below the disorder threshold of 0.5. Overall, the known target lysines interrogated in this study are in structured regions of the respective protein with three pupylation sites being in regions reaching the threshold toward disorder. Nevertheless, in terms of accessibility these three sites are not the residues with the highest accessibility.

We added our findings to the results section and included an additional Supplementary Figure (new Suppl. Fig. 2)

C4: The authors present compelling indirect evidence that charged residues are key for certain PafA:substrate interactions. However, it would be nice to see some direct structural assessment of this. I understand that the determination of the complex structure might not be feasible, but it should be possible to build a structural model of the complex using some docking method or AlphaFoldMultimer, especially so given that both partners are stable, well-folded proteins (disordered protein docking is much more difficult).

A4: We agree that a complex structure would be great to support our biochemical characterization. We tried to determine a complex structure by cryoEM but could not reach sufficient resolution to assign an orientation for PafA. We encountered severe issues in preferential orientation of the chosen substrate as well as instability of PafA at the air-water interface. Since we are limited in model substrates to reach the minimal size to make cryoEM feasible, the endeavor to solve the complex structure of PafA:substrate requires substantially more optimization which is beyond the scope of this study. In an attempt to get a model of the complex we performed AlphaFoldMultimer predictions as suggested by the reviewer. Due to limitations of the ColabFold web server we were limited to smaller monomeric substrates (FabD, PckA, Ino1). These predictions did not result in a complex but rather substrate was

placed adjacent to PafA in random orientations. This might not be surprising as the PafA:substrate interaction is transient with a determined K_d of 14 μM for PafA and PanB. Other substrates were shown to be in a similar range. We performed docking experiments using the HADDOCK 2.4 web server which allows for data-guided docking with data from other experiments. Substrates are docked into the active site with negatively charged residues coming into close proximity of the described site 2 and site 3 residues. However, docking models suggest a free 180° rotational freedom of the substrate on top of PafA. For some residues of site 2 this was previously observed (Regev et al., "An Extended Loop of the Pup Ligase, PafA, Mediates Interaction with Protein Targets" J Mol Biol 2016). Some docking results show K173 not close enough to the active site to become pupylated which does not agree with experimental results that showed K173 to be the primary target lysine (Barandun et al., "Prokaryotic ubiquitin-like protein remains intrinsically disordered when covalently attached to proteasomal target proteins" BMC Struct Biol, 2017). It is possible that the identified charge-charge interactions are most important in the initial encounter complex that is not easily predicted by these approaches. Given the very speculative nature, we do not feel confident including any docking experiments in this manuscript.

Minor points:

C5: many sentences from the first results section would fit better in the introduction, especially the ones with references

A5: We moved the introductory sentences from the "Results" section to the "Introduction".

C6: on figure 1b, the authors assess the possible influence of molecular weight on pupylation state and do not find a significant connection. First, it might be better to use the Kolmogorov-Smirnov test to test the difference between the two distributions. Second, since the rationale behind looking at molecular weight is that "larger proteins harbor more lysine residues on average", why not use the number of surface accessible lysines directly?

A6: We agree with the reviewer that "protein size" is a coarse proxy for other biologically relevant attributes. However, we chose to use protein size instead of the number of surface-accessible lysines for several reasons. First, protein size is unambiguously known for all protein sequences, whereas estimating surface accessibility requires structural information that is not available for all *Mycobacterium tuberculosis* (Mtb) proteins. Moreover, we would need to speculate on the accessibility of a lysine for the catalytic site of PafA which is not trivial as the docking experiment described above demonstrates. Second, as we mention in the manuscript, protein size can affect the likelihood of detecting a tryptic fragment, because size correlates with the number of proteolytic fragments that can be generated. Third, previous research has suggested that PafA may preferentially tag large proteins (Elharar et al. "Posttranslational regulation of coordinated enzyme activities in the Pup-proteasome system", PNAS 2016), and we wanted to examine whether this observation was reflected in the pupylome distributions. However, to get around the problem associated with estimating accessibility we included an additional panel in Supplementary Figure 1 (1c) to compare the total lysine count and the lysine sequence fraction within the pupylomes to see if pupylation substrates present on average more lysine residues. We could observe a 1.5-fold higher average lysine count and 1.2-fold higher lysine

density, overlapping with 1.2-fold larger average size for pupylated proteins. We thus feel confident that “protein size” captures similar information as “lysine number”.

As suggested by the reviewer we now include two sample Kolmogorov-Smirnov test statistics in the manuscript.

C7: in the results, the authors state that they modified the substrates by S-carboxyamidomethylation, but in figure S1c, the modification is defined as alkylation. While this is a really clever indirect way of testing for the role of structural integrity, having a degron motif would be an alternative interpretation if the modified residue happens to be inside the degron, which would also impair binding, albeit not for structural reasons.

A7: The reviewer is right that in the case of degron-guided recognition, the alteration of a cysteine within a degron by S-carboxyamidomethylation might lead to altered recognition of the substrate. However, as pointed out in A1, we do not see any indication that PafA recognizes substrates via the classical definition of a degron. In the case of FabD, two of the three cysteine residues that can be modified are internal and not surface accessible in the folded state. Instead, modification of cysteines leads to perturbation of the overall fold of FabD which in turn changes the surrounding environment of the pupylation site.

We changed out the figure labels containing “alkylation” to “mod” (Fig. 2b and Suppl. Fig. 1d and 1e) and defined this as S-carboxyamidomethylation throughout the manuscript.

C8: some alanine mutations around the pupylated lysine severely decrease pupylation efficiency. Did the authors check (e.g., with CD, which they use in other parts of the paper) these proteins to make sure the protein has the same fold?

A8: We agree with the reviewer that this is an important control to ensure that the FabD variants are properly folded. We recorded CD spectra for the four FabD variants that showed the strongest reduction of pupylation (R145A, D150A, E176A and D177A). The results are depicted in Suppl. Fig. 4c and show

that the CD spectra of the FabD variants and FabD overlay nicely, indicating that the secondary structure content has not changed.

C9: the authors identify key charged residues important for the interaction; however, based on the structures, several hydrophobic residues are close to the target lysine as well, and the importance of possible hydrophobic interactions in the PafA:substrate binding would not necessarily show up in an alanine scanning experiment. Maybe the authors could mention that hydrophobics might also play important roles in the binding event (which could be more directly assessed using e.g., FoldX on a model complex structure and checking whether the experiments and computationally calculated $\Delta\Delta G$ values are in line). This might also provide the basis for claims that the authors already make, e.g., “only a few well-positioned residues are essential for efficient pupylation”.

A9: It is definitely possible that additional hydrophobic interactions contribute to the stability of the PafA:substrate complex. Identifying these additional interactions is challenging in absence of a complex structure in sufficient resolution or a high confidence model of the complex. With the data at hand, we do not want to claim additional hydrophobic contacts we have not identified or biochemically validated. However, we now mention this possibility in the discussion. “Furthermore, it remains possible that additional residues not identified in this study also support binding of certain substrates.”

C10: what is SMR5 in figure 5b?

A10: SMR5 is the used parental Msm strain to create the *pafA* knockout strain. We changed label to parental and edited the figure legend accordingly to make this point more clear.

C11: I feel that some claims in the discussion are a bit too broad and might benefit from more careful wording. E.g. “PafA mainly utilizes site 2 arginine residues for substrate recognition, in particular R207...” this is probably only true for some substrates, while there might be others that bind with a different binding mode (especially if they are disordered).”

A11: With the complete loss of the pupylome by introduction of the R193A mutation in Msm (corresponding to R207A in Mtb) (see Figure 5b in this manuscript) we believe that this particular residue is indeed crucial for interaction with most substrates we can detect by western blotting of the lysate. The two flanking residues within site 2 also showed a strong reduction of the overall pupylome. We cannot at this point distinguish whether their role is to support R207 (e.g. orientation, formation of

charge-charge cluster) or whether they contribute directly to substrate recognition. However, they show a larger degree of involvement in comparison to residues within site 3.

Reviewer #2 (Remarks to the Author):

The manuscript "Electrostatic interactions guide substrate recognition of the prokaryotic ubiquitin-like protein ligase PafA" by Block et al is a very interesting and important study, because there is a general lack of understanding how these modified protein substrates, which are thereby marked for degradation, are recognized and selected for pupylation.

To investigate the molecular basis of pupylation by PafA the authors revisited and analyzed massspec studies on the pupylome, identified and classified substrate proteins in combination with in vitro pupylation experiments (Fig 1&2). With further biochemical and bioinformatic analysis with structure models (also using alpha-fold structural models) they determined that the substrate recognition mechanism is based on the surface located accessible specific Lysine residue in combination with charged patches on the surrounding surface area using structure function analysis with protein variants both on the substrate FabD (Fig 3) and the PafA side (Fig 4). Similar PafA variants had a clear in vivo phenotype displaying lowered to no pupylation and a defect in DNA repair comparable to a ApafA phenotype (Fig 5). Finally in one experiment they could switch the charge on one of the cognate patches on PafA and FabF reconstituting the pupylation activity of the PafA(site2) with the FabD(R2) substrate (Fig 6).

The identified cognate patches on PafA were not all equally important for the recognition of different tested substrate proteins, which might be the basis for different recognition patterns, mediating the recognition and substrate selection.

Comments and remarks:

C1: The pupylation of substrate proteins is probably mediated by the activities of the modifying PafA and the de-modifying Dop Enzyme.

Since PafA and Dop are homologs, an apparent question, which could be discussed, is whether the substrate recognition mechanism of Dop could be analogous to the described substrate interaction and recognition mechanism of PafA with substrates like FabD.

A1: This is an interesting question and we have extended the discussion to address the differences in substrate recognition between PafA and Dop. While PafA and Dop share extensive structural homology, Dop does not have to rely on recognition of the substrate itself, since it can make use of the Pup-modification as recognition determinant. Indeed, the interaction between Dop and Pup is by an order of magnitude stronger than the interaction between PafA and Pup (Özcelik et al., *Structures of Pup ligase PafA and depupylase Dop from the prokaryotic ubiquitin-like modification pathway*, Nat Commun, 2012). There are other differences between PafA and Dop apart from the substrate recognition mode, for example the orientation of the loop containing R207 (R227 in Dop). R207 points away from the active

site and was also not shown to contribute to transition state stabilization like R227 in Dop. Mutation of R227 in Dop renders it depupylation incompetent as shown previously (Cui et al., *Structures of prokaryotic ubiquitin-like protein Pup in complex with depupylase Dop reveal the mechanism of catalytic phosphate formation*, Nat Commun, 2021).

We added this point to the discussion: “While PafA and Dop are close structural homologues, by virtue of their opposing roles they do not require the same mode of substrate recognition. Dop recognizes substrates mostly through the Pup-tag and binds Pup more tightly than does PafA (Özcelik et al., *Structures of Pup ligase PafA and depupylase Dop from the prokaryotic ubiquitin-like modification pathway*, Nat Commun, 2012). R207 (R227 in Dop) is conserved between the two and is essential for Dop activity. It was shown to take part in stabilizing the phospho-Pup intermediate during the depupylation reaction (Cui et al., *Structures of prokaryotic ubiquitin-like protein Pup in complex with depupylase Dop reveal the mechanism of catalytic phosphate formation*, Nat Commun, 2021). In contrast, the PafA R207A variant is able to form the phospho-Pup species, indicating R207 does not have the same role in PafA as in Dop. Interestingly, a study converting Dop into a Pup ligase demonstrated that the function of R227 in Dop can be altered to substrate recognition by mutation of nine residues preceding R227 (Hecht et al., *Exploring Protein Space: From Hydrolase to Ligase by Substitution*, Mol Biol Evol, 2021).”

Minor comments:

C2: “scatter” should be “circle”?

–p 6 l151 “larger scatters” (Fig 1c) The term “scatter” should be “circle”?

–Fig 1 p 31 l 836 “scatter size” should be “circle size” or “size of the circle”?

–p8 l180 & 181 “filled scatter” should be “filled circle”, “open scatter” should be “open circle”?

– p7 l 155-6 “circle” instead of “scatters”?

A2: We corrected the term.

C3: Fig 1 PanB is shown in Fig 1a and 1c. Where are the in vitro substrates Adk, Mdh or ClpP2 (depicted in Fig1a) located in Fig 1c?

A3: In Fig 1c we highlight only substrates with known *in vivo* effects in lime green with the according label. Including all labels of data shown in Fig. 1c would make it unreadable and very crowded hence we left out all labels for substrates without a known *in vivo* effect. The substrates Adk, Mdh and ClpP2 fall into this latter category.

C4: Fig 1 Fig legend p 31 l 827 the reference of the mass spectrometry study is not “1”

A4: The error was fixed and now the correct reference is shown in the legend of Fig. 1.

C5: Fig 4 site 3 aa residues on PafA are depicted in 4A but no data is shown in 4b or c. Maybe somehow depict the data shown in suppl Fig 4a in Fig b or c?

A5: In Fig 5a (old 4a) the two residues making up site 3 (R442 and K444) are labeled and their effect upon mutagenesis on the pupylation of FabD and PanB is shown in Fig 4c. The assay for site 1 residues is shown in Suppl. Fig. 6a (old 5a). In Suppl. Fig 5a a comparison of pupylation of FabD between wt PafA

and the PafA3KR mutant is shown. Unless we have misunderstood the question, we believe we are presenting all the requested data.

C6: Text p7 | 155-6 “In contrast, there is a strong correlation between reactivity and steady-state accumulation in vivo (lime green scatters) in pupylation-deficient strains...”
Is the “reactivity” actually “in vitro reactivity”?

A6: Yes, the text passage refers to the *in vitro* reactivity and we have now specified that in the text.

C7: Text p7 | 158-161 These two sentences (“taken together.... pupylated by PafA”) appear to be redundant both with the text before and with the following text starting the new paragraph. Maybe one could just delete these two sentences here?

A7: We revised the paragraph to remove the redundancy.